# Learning Neural Networks with Adaptive Regularization

**Han Zhao**[*†]**, Yao-Hung Hubert Tsai**[*†]**, Ruslan Salakhutdinov**[†]**, Geoffrey J. Gordon**[†‡]
[†]Carnegie Mellon University, [‡]Microsoft Research Montreal
{han.zhao,yaohungt,rsalakhu}@cs.cmu.edu
geoff.gordon@microsoft.com

## Abstract

Feed-forward neural networks can be understood as a combination of an intermediate representation and a linear hypothesis. While most previous works aim to diversify the representations, we explore the complementary direction by performing an adaptive and data-dependent regularization motivated by the empirical Bayes method. Specifically, we propose to construct a matrix-variate normal prior (on weights) whose covariance matrix has a Kronecker product structure. This structure is designed to capture the correlations in neurons through backpropagation. Under the assumption of this Kronecker factorization, the prior encourages neurons to borrow statistical strength from one another. Hence, it leads to an adaptive and data-dependent regularization when training networks on small datasets. To optimize the model, we present an efficient block coordinate descent algorithm with analytical solutions. Empirically, we demonstrate that the proposed method helps networks converge to local optima with smaller stable ranks and spectral norms. These properties suggest better generalizations and we present empirical results to support this expectation. We also verify the effectiveness of the approach on multiclass classification and multitask regression problems with various network structures. Our code is publicly available at: https://github.com/yaohungt/Adaptive-Regularization-Neural-Network.

## 1 Introduction

Although deep neural networks have been widely applied in various domains [19, 25, 27], usually its parameters are learned via the principle of maximum likelihood, hence its success crucially hinges on the availability of large scale datasets. When training rich models on small datasets, explicit regularization techniques are crucial to alleviate overfitting. Previous works have explored various regularization [39] and data augmentation [19, 38] techniques to learn diversified representations. In this paper, we look into an alternative direction by proposing an adaptive and data-dependent regularization method to encourage neurons of the same layer to share statistical strength through exploiting correlations between data and gradients. The goal of our method is to prevent overfitting when training (large) networks on small dataset. Our key insight stems from the famous argument by Efron [8] in the literature of the empirical Bayes method: *It is beneficial to learn from the experience of others*. The empirical Bayes methods provide us a guiding principle to learn model parameters even if we do not have complete information about prior distribution. From an algorithmic perspective, we argue that the connection weights of neurons in the same layer (row/column vectors of the weight matrix) will be correlated with each other through the backpropagation learning. Hence, by learning the correlations of the weight matrix, a neuron can "borrow statistical strength" from other neurons in the same layer, which essentially increases the effective sample size during learning.

---

[*]Equal contribution.

As an illustrating example, consider a simple setting where the input $\mathbf{x} \in \mathbb{R}^d$ is fully connected to a hidden layer $\mathbf{h} \in \mathbb{R}^p$, which is further fully connected to the single output $\hat{y} \in \mathbb{R}$. Let $\sigma(\cdot)$ be the nonlinear activation function, e.g., ReLU [33], $W \in \mathbb{R}^{p \times d}$ be the connection matrix between the input layer and the hidden layer, and $\mathbf{a} \in \mathbb{R}^p$ be the vector connecting the output and the hidden layer. Without loss of generality, ignoring the bias term in each layer, we have: $\hat{y} = \mathbf{a}^T \mathbf{h}, \mathbf{h} = \sigma(W\mathbf{x})$. Consider using the usual $\ell_2$ loss function $\ell(\hat{y}, y) = \frac{1}{2}|\hat{y} - y|^2$ and take the derivative of $\ell(\hat{y}, y)$ w.r.t. $W$. We obtain the update formula in backpropagation as $W \leftarrow W - \alpha(\hat{y} - y)(\mathbf{a} \circ \mathbf{h}') \mathbf{x}^T$, where $\mathbf{h}'$ is the component-wise derivative of $\mathbf{h}$ w.r.t. its input argument, and $\alpha > 0$ is the learning rate. Realize that $(\mathbf{a} \circ \mathbf{h}') \mathbf{x}^T$ is a rank 1 matrix, and the component of $\mathbf{h}'$ is either 0 or 1. Hence, the update for each row vector of $W$ is linearly proportional to $\mathbf{x}$. Similar observation also holds for each column vector of $W$, so it implies that the row/column vectors of $W$ are correlated with each other through learning. Although in this example we only discuss a one-hidden-layer network, it is straightforward to verify that the gradient update formula for general feed-forward networks admits the same rank one structure. The above observation leads us to the following question:

> *Can we define a prior distribution over $W$ that captures the correlations through the learning process for better generalization?*

**Our Contributions**    To answer the above question, we develop an adaptive regularization method for neural nets inspired by the empirical Bayes method. Motivated by the example above, we propose a matrix-variate normal prior whose covariance matrix admits a Kronecker product structure to capture the correlations between different neurons. Using tools from convex analysis, we present an efficient block coordinate descent algorithm with closed-form solutions to optimize the model. Empirically, we show the proposed method helps the network converge to local optima with smaller stable ranks and spectral norms, and we verify the effectiveness of the approach on both multiclass classification and multitask regression problems with various network structures.

## 2    Preliminary

**Notation and Setup**    We use lowercase letter to represent scalar and lowercase bold letter to denote vector. Capital letter, e.g., $X$, is reserved for matrix. Calligraphic letter, such as $\mathcal{D}$, is used to denote set. We write $\mathrm{Tr}(A)$ as the trace of a matrix $A$, $\det(A)$ as the determinant of $A$ and $\mathrm{vec}(A)$ as $A$'s vectorization by column. $[n]$ is used to represent the set $\{1, \ldots, n\}$ for any integer $n$. Other notations will be introduced whenever needed. Suppose we have access to a training set $\mathcal{D}$ of $n$ pairs of data instances $(\mathbf{x}_i, y_i), i \in [n]$. We consider the supervised learning setting where $\mathbf{x}_i \in \mathcal{X} \subseteq \mathbb{R}^d$ and $y_i \in \mathcal{Y}$. Let $p(y \mid \mathbf{x}, \mathbf{w})$ be the conditional distribution of $y$ given $\mathbf{x}$ with parameter $\mathbf{w}$. The parametric form of the conditional distribution is assumed be known. In this paper, we assume the model parameter $\mathbf{w}$ is sampled from a prior distribution $p(\mathbf{w} \mid \theta)$ with hyperparameter $\theta$. On the other hand, given $\mathcal{D}$, the posterior distribution of $\mathbf{w}$ is denoted by $p(\mathbf{w} \mid \mathcal{D}, \theta)$.

**The Empirical Bayes Method**    To compute the predictive distribution, we need access to the value of the hyperparameter $\theta$. However, complete information about the hyperparameter $\theta$ is usually not available in practice. To this end, empirical Bayes method [1, 9, 10, 12, 36] proposes to estimate $\theta$ from the data directly using the marginal distribution:

$$\hat{\theta} = \arg\max_{\theta} \ p(\mathcal{D} \mid \theta) = \arg\max_{\theta} \int p(\mathcal{D} \mid \mathbf{w}) \cdot p(\mathbf{w} \mid \theta) \, d\mathbf{w}. \tag{1}$$

Under specific choice of the likelihood function $p(\mathbf{x}, y \mid \mathbf{w})$ and the prior distribution $p(\mathbf{w} \mid \theta)$, e.g., conjugate pairs, we can solve the above integral in closed form. In certain cases we can even obtain an analytic solution of $\hat{\theta}$, which can then be plugged into the prior distribution. At a high level, by learning the hyperparameter $\theta$ in the prior distribution directly from data, the empirical Bayes method provides us a principled and data-dependent way to obtain an estimator of $\mathbf{w}$. In fact, when both the prior and the likelihood functions are normal, it has been formally shown that the empirical Bayes estimators, e.g., the James-Stein estimator [23] and the Efron-Morris estimator [11], dominate the classic maximum likelihood estimator (MLE) in terms of quadratic loss for every choice of the model parameter $\mathbf{w}$. At a colloquial level, the success of the empirical Bayes method can be attributed to the effect of *"borrowing statistical strength"* [8], which also makes it a powerful tool in multitask learning [28, 43] and meta-learning [15].

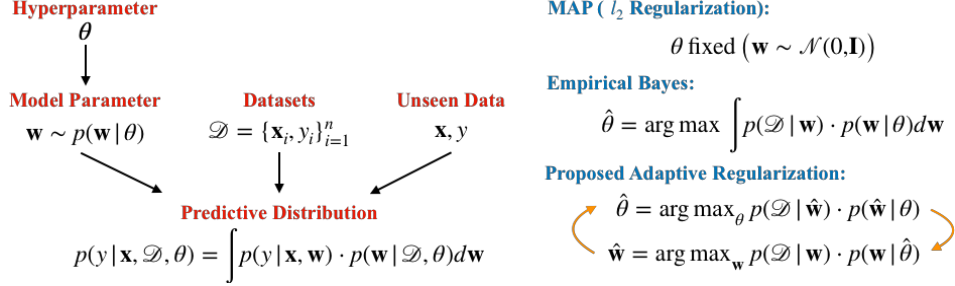

Figure 1: Illustration for Bayes/ Empirical Bayes, and our proposed adaptive regularization.

## 3 Learning with Adaptive Regularization

In this section we first propose an adaptive regularization (AdaReg) method, which is inspired by the empirical Bayes method, for learning neural networks. We then combine our observation in Sec. 1 to develop an efficient adaptive learning algorithm with matrix-variate normal prior. Through our derivation, we provide several connections and interpretations with other learning paradigms.

### 3.1 The Proposed Adaptive Regularization

When the likelihood function $p(\mathcal{D} \mid \mathbf{w})$ is implemented as a neural network, the marginalization in (1) over model parameter $\mathbf{w}$ cannot be computed exactly. Nevertheless, instead of performing expensive Monte-Carlo simulation, we propose to estimate both the model parameter $\mathbf{w}$ and the hyperparameter $\theta$ in the prior simultaneously from the joint distribution $p(\mathcal{D}, \mathbf{w} \mid \theta) = p(\mathcal{D} \mid \mathbf{w}) \cdot p(\mathbf{w} \mid \theta)$. Specifically, given an estimate $\hat{\mathbf{w}}$ of the model parameter, by maximizing the joint distribution w.r.t. $\theta$, we can obtain $\hat{\theta}$ as an approximation of the maximum marginal likelihood estimator. As a result, we can use $\hat{\theta}$ to further refine the estimate $\hat{\mathbf{w}}$ by maximizing the posterior distribution as follows:

$$\hat{\mathbf{w}} \leftarrow \max_{\mathbf{w}} \ p(\mathbf{w} \mid \mathcal{D}) = \max_{\mathbf{w}} \ p(\mathcal{D} \mid \mathbf{w}) \cdot p(\mathbf{w} \mid \hat{\theta}). \tag{2}$$

The maximizer of (2) can in turn be used in an updated joint distribution. Formally, we can define the following optimization problem that characterizes our Adaptive Regularization (AdaReg) framework:

$$\max_{\mathbf{w}} \max_{\theta} \ \log p(\mathcal{D} \mid \mathbf{w}) + \log p(\mathbf{w} \mid \theta). \tag{3}$$

It is worth connecting the optimization problem (3) to the classic maximum a posteriori (MAP) inference and also discuss their difference. If we drop the inner optimization over the hyperparameter $\theta$ in the prior distribution. Then for any fixed value $\hat{\theta}$, (3) reduces to MAP with the prior defined by the specific choice of $\hat{\theta}$, and the maximizer $\hat{\mathbf{w}}$ corresponds to the mode of the posterior distribution given by $\hat{\theta}$. From this perspective, the optimization problem in (3) actually defines a series of MAP inference problems, and the sequence $\{\hat{\mathbf{w}}_j(\hat{\theta}_j)\}_j$ defines a solution path towards the final model parameter. On the algorithmic side, the optimization problem (3) also suggests a natural block coordinate descent algorithm where we alternatively optimize over $\mathbf{w}$ and $\theta$ until the convergence of the objective function. An illustration of the framework is shown in Fig. 1.

### 3.2 Neural Network with Matrix-Normal Prior

Inspired by the observation from Sec. 1, we propose to define a matrix-variate normal distribution [16] over the connection weight matrix $W$: $W \sim \mathcal{MN}(0_{p \times d}, \Sigma_r, \Sigma_c)$, where $\Sigma_r \in \mathbb{S}^p_{++}$ and $\Sigma_c \in \mathbb{S}^d_{++}$ are the row and column covariance matrices, respectively.[2] Equivalently, one can understand the matrix-variate normal distribution over $W$ as a multivariate normal distribution with a Kronecker product covariance structure over $\text{vec}(W)$: $\text{vec}(W) \sim \mathcal{N}(0_{p \times d}, \Sigma_c \otimes \Sigma_r)$. It is then easy to check that the marginal prior distributions over the row and column vectors of $W$ are given by:

$$W_{i:} \sim \mathcal{N}(\mathbf{0}_d, [\Sigma_r]_{ii} \cdot \Sigma_c), \quad W_{:j} \sim \mathcal{N}(\mathbf{0}_p, [\Sigma_c]_{jj} \cdot \Sigma_r).$$

We point out that the Kronecker product structure of the covariance matrix exactly captures our prior about the connection matrix $W$: the fan-in/fan-out of neurons in the same layer (row/column vectors of $W$) are correlated with the same correlation matrix in the prior, and they only differ at the scales.

For illustration purpose, let us consider the simple feed-forward network discussed in Sec. 1. Consider a reparametrization of the model by defining $\Omega_r := \Sigma_r^{-1}$ and $\Omega_c := \Sigma_c^{-1}$ to be the corresponding precision matrices and plug in the prior distribution into the our AdaReg framework (see (3)). After routine algebraic simplifications, we reach the following concrete optimization problem:

$$\min_{W, \mathbf{a}} \min_{\Omega_r, \Omega_c} \quad \frac{1}{2n} \sum_{i \in [n]} (\hat{y}(\mathbf{x}_i; W, \mathbf{a}) - y_i)^2 + \lambda ||\Omega_r^{1/2} W \Omega_c^{1/2}||_F^2 - \lambda \big( d \log \det(\Omega_r) + p \log \det(\Omega_c) \big)$$

$$\text{subject to} \quad u I_p \preceq \Omega_r \preceq v I_p, \ u I_d \preceq \Omega_c \preceq v I_d \tag{4}$$

where $\lambda$ is a constant that only depends on $p$ and $d$, $0 < u \leq v$ and $uv = 1$. Note that the constraint is necessary to guarantee the feasible set to be compact so that the optimization problem is well formulated and a minimum is attainable. [3] It is not hard to show that in general the optimization problem (4) is not jointly convex in terms of $\{\mathbf{a}, W, \Omega_r, \Omega_c\}$, and this holds even if the activation function is linear. However, as we will show later, for any fixed $\mathbf{a}, W$, the reparametrization makes the partial optimization over $\Omega_r$ and $\Omega_c$ bi-convex. More importantly, we can derive an efficient algorithm that finds the optimal $\Omega_r(\Omega_c)$ for any fixed $\mathbf{a}, W, \Omega_c(\Omega_r)$ in $O(\max\{d^3, p^3\})$ time with closed form solutions. This allows us to apply our algorithm to networks of large sizes, where a typical hidden layer can contain thousands of nodes. Note that this is in contrast to solving a general semi-definite programming (SDP) problem using black-box algorithm, e.g., the interior-point method [32], which is computationally intensive and hard to scale to networks with moderate sizes. Before we delve into the details on solving (4), it is instructive to discuss some of its connections and differences to other learning paradigms.

**Maximum-A-Posteriori Estimation**. Essentially, for model parameter $W$, (4) defines a sequence of MAP problems where each MAP is indexed by the pair of precision matrices $(\Omega_r^{(t)}, \Omega_c^{(t)})$ at iteration $t$. Equivalently, at each stage of the optimization, we can interpret (4) as placing a matrix variate normal prior on $W$ where the precision matrix in the prior is given by $\Omega_r^{(t)} \otimes \Omega_c^{(t)}$. From this perspective, if we fix $\Omega_r^{(t)} = I_p$ and $\Omega_c^{(t)} = I_d$, $\forall t$, then (4) naturally reduces to learning with $\ell_2$ regularization [26]. More generally, for non-diagonal precision matrices, the regularization term for $W$ becomes:

$$||\Omega_r^{1/2} W \Omega_c^{1/2}||_F^2 = ||\text{vec}(\Omega_r^{1/2} W \Omega_c^{1/2})||_2^2 = ||(\Omega_c^{1/2} \otimes \Omega_r^{1/2}) \text{vec}(W)||_2^2,$$

and this is exactly the Tikhonov regularization [13] imposed on $W$ where the Tikhonov matrix $\Gamma$ is given by $\Gamma := \Omega_c^{1/2} \otimes \Omega_r^{1/2}$. But instead of manually designing the regularization matrix $\Gamma$ to improve the conditioning of the estimation problem, we propose to also learn both precision matrices (so $\Gamma$ as well) from data. From an algorithmic perspective, $\Gamma^T \Gamma = \Omega_c \otimes \Omega_r$ serves as a preconditioning matrix w.r.t. model parameter $W$ to reshape the gradient according to the geometry of the data [7, 17, 18].

**Volume Minimization**. Let us consider the $\log \det(\cdot)$ function over the positive definite cone. It is well known that the log-determinant function is concave [3]. Hence for any pair of matrices $A_1, A_2 \in \mathbb{S}_{++}^m$, the following inequality holds:

$$\log \det(A_1) \leq \log \det(A_2) + \langle \nabla \log \det(A_2), A_1 - A_2 \rangle = \log \det(A_2) + \text{Tr}(A_2^{-1} A_1) - m. \tag{5}$$

Applying the above inequality twice by fixing $A_1 = W \Omega_c W^T / 2d$, $A_2 = \Sigma_r$ and $A_1 = W^T \Omega_r W / 2p$, $A_2 = \Sigma_c$ respectively leads to the following inequalities:

$$d \log \det(W \Omega_c W^T / 2d) \leq -d \log \det(\Omega_r) + \frac{1}{2} \text{Tr}(\Omega_r W \Omega_c W^T) - dp,$$

$$p \log \det(W^T \Omega_r W / 2p) \leq -p \log \det(\Omega_c) + \frac{1}{2} \text{Tr}(\Omega_r W \Omega_c W^T) - dp.$$

Realize $\text{Tr}(\Omega_r W \Omega_c W^T) = ||\Omega_r^{1/2} W \Omega_c^{1/2}||_F^2$. Summing the above two inequalities leads to:

$$d \log \det(W \Omega_c W^T) + p \log \det(W^T \Omega_r W) \leq ||\Omega_r^{1/2} W \Omega_c^{1/2}||_F^2 - \big( d \log \det(\Omega_r) + p \log \det(\Omega_c) \big) + c, \tag{6}$$

where $c$ is a constant that only depends on $d$ and $p$. Recall that $|\det(A^T A)|$ computes the squared volume of the parallelepiped spanned by the column vectors of $A$. Hence (6) gives us a natural

interpretation of the objective function in (4): the regularizer essentially upper bounds the log-volume of the two parallelpipeds spanned by the row and column vectors of $W$. But instead of measuring the volume using standard Euclidean inner product, it also takes into account the local curvatures defined by $\Sigma_r$ and $\Sigma_c$, respectively. For vectors with fixed lengths, the volume of the parallelepiped spanned by them becomes smaller when they are more linearly correlated, either positively or negatively. At a colloquial level, this means that the regularizer in (4) forces fan-in/fan-out of neurons at the same layer to be either positively or negatively correlated with each other, and this corresponds exactly to the effect of sharing statistical strengths.

### 3.3 The Algorithm

In this section we describe a block coordinate descent algorithm to optimize the objective function in (4) and detail how to efficiently solve the matrix optimization subproblems in closed form using tools from convex analysis. Due to space limit, we defer proofs and detailed derivation to appendix. Given a pair of constants $0 < u \leq v$, we define the following thresholding function $\mathbb{T}_{[u,v]}(x)$:

$$\mathbb{T}_{[u,v]}(x) := \max\{u, \min\{v, x\}\}. \tag{7}$$

We summarize our block coordinate descent algorithm to solve (4) in Alg. 1. In each iteration, Alg. 1 takes a first-order algorithm $\mathfrak{A}$, e.g., the stochastic gradient descent, to optimize the parameters of the neural network by backpropagation. It then proceeds to compute the optimal solutions for $\Omega_r$ and $\Omega_c$ using INVTHRESHOLD as a sub-procedure. Alg. 1 terminates when a stationary point is found.

We now proceed to show that the procedure INVTHRESHOLD finds the optimal solution given all the other variables fixed. Due to the symmetry between $\Omega_r$ and $\Omega_c$ in (4), we will only prove this for $\Omega_r$, and similar arguments can be applied to $\Omega_c$ as well. Fix both $W$, $\Omega_c$ and ignore all the terms that do not depend on $\Omega_r$, the sub-problem on optimizing $\Omega_r$ becomes:

$$\min_{\Omega_r} \quad \mathrm{Tr}(\Omega_r W \Omega_c W^T) - d \log \det(\Omega_r), \qquad \text{subject to} \quad u I_p \preceq \Omega_r \preceq v I_p. \tag{8}$$

It is not hard to show that the optimization problem (9) is convex. Define the constraint set $\mathcal{C} := \{A \in \mathbb{S}^p_{++} \mid u I_p \preceq A \preceq v I_p\}$ and the indicator function $\mathbb{I}_{\mathcal{C}}(A) = 0$ iff $A \in \mathcal{C}$ else $\infty$. Given the convexity of (9), we can use the indicator function to first transform (9) into the following unconstrained one:

$$\min_{\Omega_r} \quad \mathrm{Tr}(\Omega_r W \Omega_c W^T) - d \log \det(\Omega_r) + \mathbb{I}_{\mathcal{C}}(\Omega_r). \tag{9}$$

Then we can use the first-order optimality condition to characterize the optimal solution:

$$0 \in \partial\left(\frac{1}{d}\mathrm{Tr}(\Omega_r W \Omega_c W^T) - \log \det(\Omega_r) + \mathbb{I}_{\mathcal{C}}(\Omega_r)\right) = W\Omega_c W^T/d - \Omega_r^{-1} + \mathcal{N}_{\mathcal{C}}(\Omega_r),$$

where $\mathcal{N}_{\mathcal{C}}(A) := \{B \in \mathbb{S}^p \mid \mathrm{Tr}(B^T(Z - A)) \leq 0, \forall Z \in \mathcal{C}\}$ is the normal cone w.r.t. $\mathcal{C}$ at $A$. The following key lemma characterizes the structure of the normal cone:

**Lemma 1.** Let $\Omega_r \in \mathcal{C}$, then $\mathcal{N}_{\mathcal{C}}(\Omega_r) = -\mathcal{N}_{\mathcal{C}}(\Omega_r^{-1})$.

Equivalently, combining Lemma 1 with the optimality condition, we have

$$W\Omega_c W^T/d - \Omega_r^{-1} \in \mathcal{N}_{\mathcal{C}}(\Omega_r^{-1}).$$

Geometrically, this means that the optimum $\Omega_r^{-1}$ is the Euclidean projection of $W\Omega_c W^T/d$ onto $\mathcal{C}$. Hence in order to solve (9), it suffices if we can solve the following Euclidean projection problem efficiently, where $\widetilde{\Omega_r} \in \mathbb{S}^p$ is a given real symmetric matrix:

$$\min_{\Omega_r} \quad ||\Omega_r - \widetilde{\Omega_r}||^2_F, \qquad \text{subject to} \quad u I_p \preceq \Omega_r \preceq v I_p. \tag{10}$$

Perhaps a little bit surprising, we can find the optimal solution to the above Euclidean projection problem efficiently in closed form:

**Theorem 1.** Let $\widetilde{\Omega_r} \in \mathbb{S}^p$ with eigendecomposition as $\widetilde{\Omega_r} = Q\Lambda Q^T$ and $\mathrm{Proj}_{\mathcal{C}}(\cdot)$ be the Euclidean projection operator onto $\mathcal{C}$, then $\mathrm{Proj}_{\mathcal{C}}(\widetilde{\Omega_r}) = Q\mathbb{T}_{[u,v]}(\Lambda)Q^T$.

**Corollary 1.** Let $W\Omega_c W^T$ be eigendecomposed as $Q\mathrm{diag}(\mathbf{r})Q^T$, then the optimal solution to (9) is given by $Q\mathbb{T}_{[u,v]}(d/\mathbf{r})Q^T$.

Similar arguments can be made to derive the solution for $\Omega_c$ in (4). The final algorithm is very simple as it only contains one SVD, hence its time complexity is $O(\max\{d^3, p^3\})$. Note that the total number of parameters in the network is at least $\Omega(dp)$, hence the algorithm is efficient as it scales sub-quadratically in terms of number of parameters in the network.

---

**Algorithm 1** Block Coordinate Descent for Adaptive Regularization

---

**Input:** Initial value $\phi^{(0)} := \{\mathbf{a}^{(0)}, W^{(0)}\}$, $\Omega_r^{(0)} \in \mathbb{S}_{++}^p$ and $\Omega_c^{(0)} \in \mathbb{S}_{++}^d$, first-order optimization algorithm $\mathfrak{A}$.

1: **for** $t = 1, \ldots, \infty$ until convergence **do**
2:      Fix $\Omega_r^{(t-1)}, \Omega_c^{(t-1)}$, optimize $\phi^{(t)}$ by backpropagation and algorithm $\mathfrak{A}$
3:      $\Omega_r^{(t)} \leftarrow \textsc{InvThreshold}(W^{(t)}\Omega_c^{(t-1)}W^{(t)T}, d, u, v)$
4:      $\Omega_c^{(t)} \leftarrow \textsc{InvThreshold}(W^{(t)T}\Omega_r^{(t)}W^{(t)}, p, u, v)$
5: **end for**
6: **procedure** $\textsc{InvThreshold}(\Delta, m, u, v)$
7:      Compute SVD: $Q\text{diag}(\mathbf{r})Q^T = \text{SVD}(\Delta)$
8:      Hard thresholding $\mathbf{r}' \leftarrow \mathbb{T}_{[u,v]}(m/\mathbf{r})$
9:      **return** $Q\text{diag}(\mathbf{r}')Q^T$
10: **end procedure**

---

# 4 Experiments

In this section we demonstrate the effectiveness of AdaReg in learning practical deep neural networks on real-world datasets. We report generalization, optimization as well as stability results.

## 4.1 Experimental Setup

**Multiclass Classification (MNIST & CIFAR10)**: In this experiment, we show that AdaReg provides an effective regularization on the network parameters. To this end, we use a convolutional neural network as our baseline model. To show the effect of regularization, we gradually increase the training set size. In MNIST we use the step from 60 to 60,000 (11 different experiments) and in CIFAR10 we consider the step from 5,000 to 50,000 (10 different experiments). For each training set size, we repeat the experiments for 10 times. The mean along with its standard deviation are shown as the statistics. Moreover, since both the optimization and generalization of neural networks are sensitive to the size of minibatches [14, 24], we study two minibatch settings for 256 and 2048, respectively. In our method, we place a matrix-variate normal prior over the weight matrix of the last softmax layer, and we use Alg. 1 to optimize both the model weights and two covariance matrices.

**Multitask Regression (SARCOS)**: SARCOS relates to an inverse dynamics problem for a seven degree-of-freedom (DOF) SARCOS anthropomorphic robot arm [41]. The goal of this task is to map from a 21-dimensional input space (7 joint positions, 7 joint velocities, 7 joint accelerations) to the corresponding 7 joint torques. Hence there are 7 tasks and the inputs are shared among all the tasks. The training set and test set contain 44,484 and 4,449 examples, respectively. Again, we apply AdaReg on the last layer weight matrix, where each row corresponds to a separate task vector.

We compare AdaReg with classic regularization methods in the literature, including weight decay, dropout [39], batch normalization (BN) [22] and the DeCov method [6]. We also note that we fix all the hyperparameters such as learning rate to be the same for all the methods. We report evaluation metrics on test set as a measure of generalization. To understand how the proposed adaptive regularization helps in optimization, we visualize the trajectory of the loss function during training. Lastly, we also present the inferred correlation of the weight matrix for qualitative study.

## 4.2 Results and Analysis

**Multiclass Classification (MNIST & CIFAR10)**: Results on the multiclass classification for different training sizes are show in Fig. 2. For both MNIST and CIFAR10, we find AdaReg, Weight Decay, and Dropout are the effective regularization methods, while Batch Normalization and DeCov vary in different settings. Batch Normalization suffers from large batch size in CIFAR10 (comparing Fig. 2 (c) and (d)) but is not sensitive to batch size in MNIST (comparing Fig. 2 (a) and (b)). The performance deterioration in large batch size of Batch Normalization is also observed by [21]. DeCov, on the other hand, improves the generalization in MNIST with batch size 256 (see Fig. 2 (a)), while it demonstrates only comparable or even worse performance in other settings. To conclude, as training set size grows, AdaReg consistently performs better generalization as comparing to other regularization methods. We also note that AdaReg is not sensitive to the size of minibatches while most of the methods suffer from large minibatches. In appendix, we show the combination of AdaReg with other generalization methods can usually lead to even better results.

Table 1: Explained variance of different methods on 7 regression tasks from the SARCOS dataset.

| Method | 1st | 2nd | 3rd | 4th | 5th | 6th | 7th |
|--------|-----|-----|-----|-----|-----|-----|-----|
| **MTL** | 0.4418 | 0.3472 | 0.5222 | 0.5036 | 0.6024 | 0.4727 | 0.5298 |
| **MTL-Dropout** | 0.4413 | 0.3271 | 0.5202 | 0.5063 | 0.6036 | 0.4711 | 0.5345 |
| **MTL-BN** | 0.4768 | 0.3770 | 0.5396 | 0.5216 | 0.6117 | 0.4936 | 0.5479 |
| **MTL-DeCoV** | 0.4027 | 0.3137 | 0.4703 | 0.4515 | 0.5229 | 0.4224 | 0.4716 |
| **MTL-AdaReg** | **0.4769** | **0.3969** | **0.5485** | **0.5308** | **0.6202** | **0.5085** | **0.5561** |

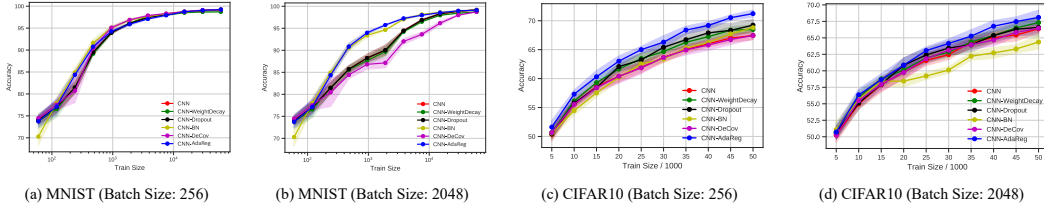

(a) MNIST (Batch Size: 256)    (b) MNIST (Batch Size: 2048)    (c) CIFAR10 (Batch Size: 256)    (d) CIFAR10 (Batch Size: 2048)

Figure 2: Generalization performance on MNIST and CIFAR10. AdaReg improves generalization under both minibatch settings.

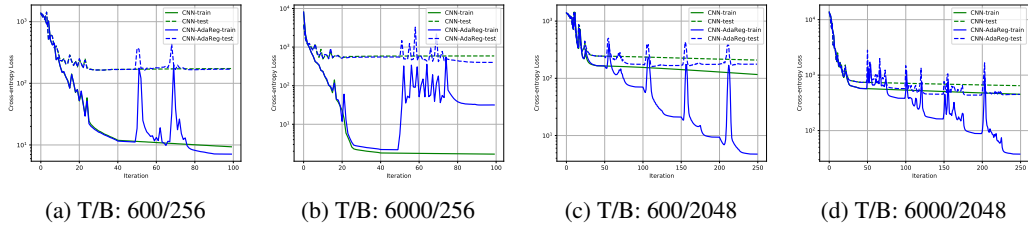

(a) T/B: 600/256    (b) T/B: 6000/256    (c) T/B: 600/2048    (d) T/B: 6000/2048

Figure 3: Optimization trajectory of AdaReg on MNIST with training size/batch size on training and test sets. AdaReg helps to converge to better local optima. Note the $\log$-scale on $y$-axis.

**Multitask Regression (SARCOS)**: In this experiment we are interested in investigating whether AdaReg can lead to better generalization for multiple related regression problems. To do so, we report the explained variance as a normalized metric, e.g., one minus the ratio between mean squared error and the variance of different methods in Table 1. The larger the explained variance, the better the predictive performance. In this case we observe a consistent improvement of AdaReg over other competitors on all the 7 regression tasks. We would like to emphasize that all the experiments share exactly the same experimental protocol, including network structure, optimization algorithm, training iteration, etc, so that the performance differences can only be explained by different ways of regularizations. For better visualization, we also plot the result in appendix.

**Optimization**: It has recently been empirically shown that BN helps optimization not by reducing internal covariate shift, but instead by smoothing the landscape of the loss function [37]. To understand how AdaReg improves generalization, in Fig. 3, we plot the values of the cross entropy loss function on both the training and test sets during optimization using Alg. 1. The experiment is performed in MNIST with batch size 256/2048. In this experiment, we fix the number of outer loop to be 2/5 and each block optimization over network weights contains 50 epochs. Because of the stochastic optimization over model weights, we can see several unstable peaks in function value around iteration 50 when trained with AdaReg, which corresponds to the transition phase between two consecutive outer loops with different row/column covariance matrices. In all the cases AdaReg converges to better local optima of the loss landscape, which lead to better generalization on the test set as well because they have smaller loss values on the test set when compared with training without AdaReg.

**Stable rank and spectral norm**: Given a matrix $W$, the stable rank of $W$, denoted as srank$(W)$, is defined as srank$(W) := ||W||_F^2 / ||W||_2^2$. As its name suggests, the stable rank is more stable than the rank because it is largely unaffected by tiny singular values. It has recently been shown [34, Theorem 1] that the generalization error of neural networks crucially depends on both the stable ranks and the spectral norms of connection matrices in the network. Specifically, it can be shown that the generalization error is upper bounded by $O\big(\sqrt{\prod_{j=1}^{L} ||W_j||_2^2 \sum_{j=1}^{L} \text{srank}(W_j)/n}\big)$, where $L$ is the

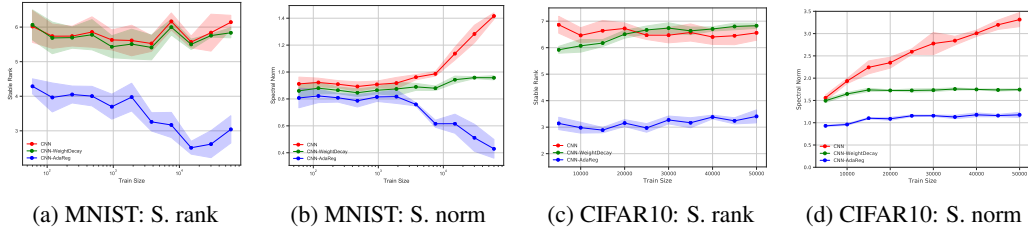

| (a) MNIST: S. rank | (b) MNIST: S. norm | (c) CIFAR10: S. rank | (d) CIFAR10: S. norm |

Figure 4: Comparisons of stable ranks (S. rank) and spectral norms (S. norm) from different methods on MNIST and CIFAR10. $x$-axis corresponds to the training size.

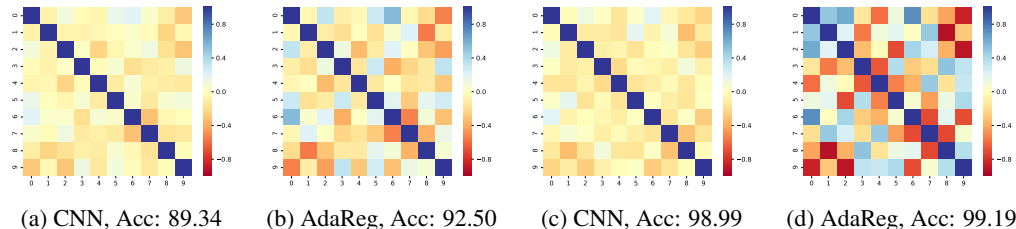

| (a) CNN, Acc: 89.34 | (b) AdaReg, Acc: 92.50 | (c) CNN, Acc: 98.99 | (d) AdaReg, Acc: 99.19 |

Figure 5: Correlation matrix of the weight matrix in the softmax layer. The left two correspond to dataset with training size 600 and the right two with size 60,000. Acc means the test set accuracy.

number of layers in the network. Essentially, this upper bound suggests that smaller spectral norm (smoother function mapping) and stable rank (skewed spectrum) leads to better generalization.

To understand why AdaReg improves generalization, in Fig. 4, we plot both the stable rank and the spectral norm of the weight matrix in the last layer of the CNNs used in our MNIST and CIFAR10 experiments. We compare 3 methods: CNN without any regularization, CNN trained with weight decay and CNN with AdaReg. For each setting we repeat the experiments for 5 times, and we plot the mean along with its standard deviation. From Fig. 4a and Fig. 4c it is clear that AdaReg leads to a significant reduction in terms of the stable rank when compared with weight decay, and this effect is consistent in all the experiments with different training size. Similarly, in Fig. 4b and Fig. 4d we plot the spectral norm of the weight matrix. Again, both weight decay and AdaReg help reduce the spectral norm in all settings, but AdaReg plays a more significant role than the usual weight decay. Combining the experiments with the generalization upper bound introduced above, we can see that training with AdaReg leads to an estimator of $W$ that has lower stable rank and smaller spectral norm, which explains why it achieves a better generalization performance.

Furthermore, this observation holds on the SARCOS datasets as well. For the SARCOS dataset, the weight matrix being regularized is of dimension $100 \times 7$. Again, we compare the results using three methods: MTL, MTL-WeightDecay and MTL-AdaReg. As can be observed from Table 2, compared with the weight decay regularization, AdaReg substantially reduces both the stable rank and the spectral norm of learned weight matrix, which also helps to explain why MTL-AdaReg generalizes better compared with MTL and MTL-WeightDecay.

Table 2: Stable rank and spectral norm on SARCOS.

|  | MTL | MTL-WeightDecay | MTL-AdaReg |
| --- | --- | --- | --- |
| **Stable Rank** | 4.48 | 4.83 | **2.88** |
| **Spectral Norm** | 0.96 | 0.92 | **0.70** |

**Correlation Matrix**: To verify that AdaReg imposes the effect of "sharing statistical strength" during training, we visualize the weight matrix of the softmax layer by computing the corresponding correlation matrix, as shown in Fig. 5. In Fig. 5, darker color means stronger correlation. We conduct two experiments with training size 600 and 60,000 respectively. As we can observe, training with AdaReg leads to weight matrix with stronger correlations, and this effect is more evident when the training set is large. This is consistent with our analysis of sharing statistical strengths. As a sanity check, from Fig. 5 we can also see that similar digits, e.g., 1 and 7, share a positive correlation while dissimilar ones, e.g., 1 and 8, share a negative correlation.

# 5 Related Work

**The Empirical Bayes Method vs Bayesian Neural Networks**   Despite the name, empirical Bayes method is in fact a frequentist approach to obtain estimator with favorable properties. On the other hand, truly Bayesian inference would instead put a posterior distribution over model weights to characterize the uncertainty during training [2, 20, 30]. However, due to the complexity of nonlinear neural networks, analytic posterior is not available, hence strong independent assumptions over model weight have to be made in order to achieve computationally tractable variational solution. Typically, both the prior and the variational posterior are assumed to fully factorize over model weights. As an exception, Louizos and Welling [29], Sun et al. [40] seek to learn Bayesian neural nets where they approximate the intractable posterior distribution using matrix-variate Gaussian distribution. The prior for weights are still assumed to be known and fixed. As a comparison, we use matrix-variate Gaussian as the prior distribution and we learn the hyperparameter in the prior from data. Hence our method does not belong to Bayesian neural nets: we instead use the empirical Bayes principle to derive adaptive regularization method in order to have better generalization, as done in [4, 35].

**Regularization Techniques in Deep Learning**   Different kinds of regularization approaches have been studied and designed for neural networks, e.g., weight decay [26], early stopping [5], Dropout [39] and the more recent DeCov [6] method. BN was proposed to reduce the internal covariate shift during training, but recently it has been empirically shown to actually smooth the landscape of the loss function [37]. As a comparison, we propose AdaReg as an adaptive regularization method, with the aim to reduce overfitting by allowing neurons to share statistical strengths. From the optimization perspective, learning the row and column covariance matrices help to converge to better local optimum that also generalizes better.

**Kronecker Factorization in Optimization**   The Kronecker factorization assumption has also been applied in the literature of neural networks to approximate the Fisher information matrix in second-order optimization methods [31, 42]. The main idea here is to approximate the curvature of the loss function's landscape, in order to achieve better convergence speed compared with first-order method while maintaining the tractability of such computation. Different from these work, here in our method we assume a Kronecker factorization structure on the covariance matrix of the prior distribution, not the Fisher information matrix of the log-likelihood function. Furthermore, we also derive closed-form solutions to optimize these factors without any kind of approximations.

# 6 Conclusion

Inspired by empirical Bayes method, in this paper we propose an adaptive regularization (AdaReg) with matrix-variate normal prior for model parameters in deep neural networks. The prior encourages neurons to borrow statistical strength from other neurons during the learning process, and it provides an effective regularization when training networks on small datasets. To optimize the model, we design an efficient block coordinate descent algorithm to learn both model weights and the covariance structures. Empirically, on three datasets we demonstrate that AdaReg improves generalization by finding better local optima with smaller spectral norms and stable ranks. We believe our work takes an important step towards exploring the combination of ideas from the empirical Bayes literature and rich prediction models like deep neural networks. One interesting direction for future work is to extend the current approach to online setting where we only have access to one training instance at a time, and to analyze the property of such method in terms of regret analysis with adaptive optimization methods.

**Acknowledgments**

HZ and GG would like to acknowledge support from the DARPA XAI project, contract #FA87501720152 and an Nvidia GPU grant. YT and RS were supported in part by DARPA grant FA875018C0150, DARPA SAGAMORE HR00111990016, Office of Naval Research grant N000141812861, AFRL CogDeCON, and Apple. YT and RS would also like to acknowledge NVIDIA's GPU support. Last, we thank Denny Wu for suggestions on exploring and analyzing our algorithm in terms of stable rank.

## Footnotes

[2]The probability density function is given by $p(W \mid \Sigma_r, \Sigma_c) = \frac{\exp\left(- \text{Tr}(\Sigma_r^{-1} W \Sigma_c^{-1} W^T)/2\right)}{(2\pi)^{pd/2} \det(\Sigma_r)^{d/2} \det(\Sigma_c)^{p/2}}.$

[3]The constraint $uv = 1$ is only for the ease of presentation in the following part and can be readily removed.

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
