[Supplementary Material · supplement.pdf]

# Learning Neural Networks with Adaptive Regularization

## Abstract

Feed-forward neural networks can be understood as a combination of an interme-
diate representation and a linear hypothesis. While most previous works aim to
diversify the representations, we explore the complementary direction by perform-
ing an adaptive and data-dependent regularization motivated by the empirical Bayes
method. Specifically, we propose to construct a matrix-variate normal prior (on
weights) whose covariance matrix has a Kronecker product structure. This structure
is designed to capture the correlations in neurons through backpropagation. Under
the assumption of this Kronecker factorization, the prior encourages neurons to
borrow statistical strength from one another. Hence, it leads to an adaptive and
data-dependent regularization when training networks on small datasets. To opti-
mize the model, we present an efficient block coordinate descent algorithm with
analytical solutions. Empirically, we demonstrate that the proposed method helps
networks converge to local optima with smaller stable ranks and spectral norms.
These properties suggest better generalizations and we present empirical results
to support this expectation. We also verify the effectiveness of the approach on
multiclass classification and multitask regression problems with various network
structures.

## 1 Introduction

Although deep neural networks have been widely applied in various domains [19, 25, 29], usually its
parameters are learned via the principle of maximum likelihood, hence its success crucially hinges
on the availability of large scale datasets. When training rich models on small datasets, explicit
regularization techniques are crucial to alleviate overfitting. Previous works have explored various
regularization [42] and data augmentation [19, 41] techniques to learn diversified representations.
In this paper, we look into an alternative direction by proposing an adaptive and data-dependent
regularization method to encourage neurons of the same layer to share statistical strength. The goal of
our method is to prevent overfitting when training (large) networks on small dataset. Our key insight
stems from the famous argument by Efron [8] in the literature of the empirical Bayes method: *It is
beneficial to learn from the experience of others*. From an algorithmic perspective, we argue that the
connection weights of neurons in the same layer (row/column vectors of the weight matrix) will be
correlated with each other through the backpropagation learning. Hence, by learning the correlations
of the weight matrix, a neuron can "borrow statistical strength" from other neurons in the same layer.

As an illustrating example, consider a simple setting where the input $\mathbf{x} \in \mathbb{R}^d$ is fully connected to a
hidden layer $\mathbf{h} \in \mathbb{R}^p$, which is further fully connected to the single output $\hat{y} \in \mathbb{R}$. Let $\sigma(\cdot)$ be the
nonlinear activation function, e.g., ReLU [36], $W \in \mathbb{R}^{p \times d}$ be the connection matrix between the
input layer and the hidden layer, and $\mathbf{a} \in \mathbb{R}^p$ be the vector connecting the output and the hidden layer.
Without loss of generality, ignoring the bias term in each layer, we have: $\hat{y} = \mathbf{a}^T \mathbf{h}, \mathbf{h} = \sigma(W\mathbf{x})$.
Consider using the usual $\ell_2$ loss function $\ell(\hat{y}, y) = \frac{1}{2}|\hat{y} - y|^2$ and take the derivative of $\ell(\hat{y}, y)$ w.r.t.

$W$. We obtain the update formula in backpropagation as $W \leftarrow W - \alpha(\hat{y} - y)(\mathbf{a} \circ \mathbf{h}') \, \mathbf{x}^T$, where $\mathbf{h}'$ is the componentwise derivative of $\mathbf{h}$ w.r.t. its input argument, and $\alpha > 0$ is the learning rate. Realize that $(\mathbf{a} \circ \mathbf{h}') \, \mathbf{x}^T$ is a rank 1 matrix, and the component of $\mathbf{h}'$ is either 0 or 1. Hence, the update for each row vector of $W$ is linearly proportional to $\mathbf{x}$. Note that the observation holds for any input pair $(\mathbf{x}, y)$, so the update formula implies that the row

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

In this appendix we first describe more related work, and then present missing proofs in the main section. We also provide detailed description of our experiments.

## A More Related Work

Different kinds of regularization approaches have been studied and designed for neural networks, e.g., weight decay [26], early stopping [5], Dropout [42] and the more recent DeCov [6] method. BN was proposed to reduce the internal covariate shift during training, but recently it has been empirically shown to actually smooth the landscape of the loss function [40]. As a comparison, we propose AdaReg as an adaptive regularization method, with the aim to reduce overfitting by allowing neurons to share statistical strengths. From the optimization perspective, learning the row and column covariance matrices help to converge to better local optimum that also generalizes better.

The Kronecker factorization assumption has also been applied in the literature of neural networks to approximate the Fisher information matrix in second-order optimization methods [34, 45]. The main idea here is to approximate the curvature of the loss function's landscape, in order to achieve better convergence speed compared with first-order method while maintaining the tractability of such computation.

Determinantal point process (DPP) has been previously applied to compress neural networks [33]. Specifically, a DPP kernel is placed over the activations of neurons from the same layer, and then neurons with similar activations over a fixed dataset are merged into a single one. However, it is well known that DPPs can capture only negative correlations [27, 28], and as a result they do not stimulate neurons to learn from the experience of other neurons. As a comparison, by explicitly learning both precision (covariance) matrices, our framework can account for both positive and negative correlations among fan-in/fan-out of neurons from the same layer.

## B Detailed Derivation and Proofs of Our Algorithm

We first show that the optimization problem (8) is convex:

**Proposition 1.** The optimization problem (8) is convex.

*Proof.* It is clear that the objective function is convex: the trace term is linear in $\Omega_r$ and it is well-known that the $\log \det(\cdot)$ is concave in the positive definite cone [3], hence it trivially follows that $\mathrm{Tr}(\Omega_r W \Omega_c W^T) - d \log \det(\Omega_r)$ is convex in $\Omega_r$.

It remains to show that the constraint set is also convex. Let $\Omega_1, \Omega_2$ be any feasible points, i.e., $uI_p \preceq \Omega_1 \preceq vI_p$ and $uI_p \preceq \Omega_2 \preceq vI_p$. Let $\forall t \in (0, 1)$, we have:

$$||t\Omega_1 + (1 - t)\Omega_2||_2 \leq t||\Omega_1||_2 + (1 - t)||\Omega_2||_2 \leq tv + (1 - t)v = v,$$

where we use $|| \cdot ||_2$ to denote the spectral norm of a matrix. Now since both $\Omega_1$ and $\Omega_2$ are positive definite, the spectral norm is also the largest eigenvalue, hence this shows that $t\Omega_1 + (1-t)\Omega_2 \preceq vI_p$.

To show the other direction, we use the Courant-Fischer characterization of eigenvalues. Let $\lambda_{\min}(A)$ denote the minimum eigenvalue of a real symmetric matrix $A$, then by the Courant-Fischer min-max theorem, we have:

$$\lambda_{\min}(A) := \min_{\mathbf{x} \neq 0, ||\mathbf{x}||_2 = 1} ||A\mathbf{x}||_2.$$

For the matrix $t\Omega_1 + (1-t)\Omega_2$, let $\mathbf{x}^*$ be the vector corresponding to the minimum eigenvalue, hence we have:

$$\begin{aligned}
\lambda_{\min}(t\Omega_1 + (1-t)\Omega_2) &= \min_{\mathbf{x} \neq 0, ||\mathbf{x}||_2 = 1} ||(t\Omega_1 + (1-t)\Omega_2)\mathbf{x}||_2 \\
&= (t\Omega_1 + (1-t)\Omega_2)\mathbf{x}^* \\
&\geq t\lambda_{\min}(\Omega_1) + (1-t)\lambda_{\min}(\Omega_2) \\
&\geq tu + (1-t)u \\
&= u,
\end{aligned}$$

which also means that $t\Omega_1 + (1-t)\Omega_2 \succeq uI_p$, and this completes the proof. ∎

450 The following key lemma characterizes the structure of the normal cone:

451 **Lemma 1.** Let $\Omega_r \in \mathcal{C}$, then $\mathcal{N}_{\mathcal{C}}(\Omega_r) = -\mathcal{N}_{\mathcal{C}}(\Omega_r^{-1})$.

452 *Proof.* Let $S \in \mathcal{N}_{\mathcal{C}}(\Omega_r)$. We want to show $-S \in \mathcal{N}_{\mathcal{C}}(\Omega_r^{-1})$. By definition of the normal cone, since
453 $S \in \mathcal{N}_{\mathcal{C}}(\Omega_r)$, we have:
$$\text{Tr}(SZ) \leq \text{Tr}(S\Omega_r), \quad \forall Z \in \mathcal{C}$$
454 Now realize that $\Omega_r \in \mathcal{C}$ and $\mathcal{C}$ is a compact set, it follows $\Omega_r$ is the solution of the following linear
455 program:
$$\max \quad \text{Tr}(SZ), \qquad \text{subject to} \quad Z \in \mathcal{C}$$
456 Since both $S$ and $Z$ are real symmetric matrix, we can decompose them as $Z := Q_Z \Lambda_Z Q_Z^T$ and
457 $S := Q_S \Lambda_S Q_S^T$, where both $Q_Z, Q_S$ are orthogonal matrices and $\Lambda_Z, \Lambda_S$ are diagonal matrices with
458 the corresponding eigenvalues in decreasing order. Plug them into the objective function, we have:
$$\text{Tr}(SZ) = \text{Tr}(Q_S \Lambda_S Q_S^T Q_Z \Lambda_Z Q_Z^T) = \text{Tr}(\Lambda_S Q_S^T Q_Z \Lambda_Z Q_Z^T Q_S).$$
459 Define $K := Q_S^T Q_Z$ and $D = K \circ K$, where we use $\circ$ to denote the Hadamard product between two
460 matrices. Since both $Q_S$ and $Q_Z$ are orthogonal matrices, we know that $K$ is also orthogonal, which
461 implies:
$$\sum_{j=1}^{p} D_{ij} = 1, \forall i \in [p], \quad \text{and} \quad \sum_{i=1}^{p} D_{ij} = 1, \forall j \in [p].$$
462 As a result, $D$ is a doubly stochastic matrix and we can further simplify the objective function as:
$$\text{Tr}(\Lambda_S Q_S^T Q_Z \Lambda_Z Q_Z^T Q_S) = \text{Tr}(\Lambda_S K \Lambda_Z K^T) = \lambda_S^T D \lambda_Z = \sum_{i,j=1}^{p} \lambda_{S,i} D_{ij} \lambda_{Z,j},$$
463 where $\lambda_S$ and $\lambda_Z$ are $p$ dimensional vectors that contain the eigenvalues of $S$ and $Z$ in decreasing
464 order, respectively. Now for any $\lambda_S$ and $\lambda_Z$ in decreasing order, we have:
$$u \sum_{i=1}^{p} \lambda_{S,i} \leq \sum_{i=1}^{p} \lambda_{S,i} \lambda_{Z,1+p-i} \leq \sum_{i,j=1}^{p} \lambda_{S,i} D_{ij} \lambda_{Z,j} \leq \sum_{i=1}^{p} \lambda_{S,i} \lambda_{Z,i} \leq v \sum_{i=1}^{p} \lambda_{S,i} \quad (10)$$
465 From (10), in order for $\Omega_r$ to maximize the linear program, it must hold that $D = K = I_p$ and all the
466 eigenvalues of $\Omega_r$ are $v$. But due to the assumption that $uv = 1$, in this case we also know that all
467 the eigenvalues of $\Omega_r^{-1}$ are $1/v = u$, hence $\Omega_r^{-1}$ also minimizes the above linear program, which
468 implies:
$$\text{Tr}(S\Omega_r^{-1}) \leq \text{Tr}(SZ), \quad \forall Z \in \mathcal{C} \Leftrightarrow \text{Tr}(-S(Z - \Omega_r^{-1})) \leq 0 \quad \forall Z \in \mathcal{C}.$$
469 In other words, we have $-S \in \mathcal{N}_{\mathcal{C}}(\Omega_r^{-1})$. Using exactly the same arguments it is clear to see that the
470 other direction also holds, hence we have $\mathcal{N}_{\mathcal{C}}(\Omega_r) = -\mathcal{N}_{\mathcal{C}}(\Omega_r^{-1})$. ∎

471 Based on the previous first-order optimality condition, it is clear to see that Lemma 1 implies
472 $W\Omega_c W^T/d - \Omega_r^{-1} \in \mathcal{N}_{\mathcal{C}}(\Omega_r^{-1})$. Geometrically, this means that the optimum $\Omega_r^{-1}$ is the Euclidean
473 projection of $W\Omega_c W^T/d$ onto $\mathcal{C}$. Hence we proceed to derive the projection operator:

474 **Theorem 1.** Let $\widetilde{\Omega_r} \in \mathbb{S}^p$ with eigendecomposition as $\widetilde{\Omega_r} = Q\Lambda Q^T$ and $\text{Proj}_{\mathcal{C}}(\cdot)$ be the Euclidean
475 projection operator onto $\mathcal{C}$, then $\text{Proj}_{\mathcal{C}}(\widetilde{\Omega_r}) = Q\mathbb{T}_{[u,v]}(\Lambda)Q^T$.

476 *Proof.* Since $\Omega_r \in \mathcal{C}$ is real and symmetric, we can reparametrize $\Omega_r$ as $\Omega_r := U\Lambda_{\Omega_r} U^T$ where $U$
477 is an orthogonal matrix and $\Lambda_{\Omega_r}$ is a diagonal matrix whose entries corresponds to the eigenvalues of
478 $\Omega_r$. Recall that $U$ corresponds to a rigid transformation that preserves length, so we have:
$$||\Omega_r - \widetilde{\Omega_r}||_F^2 = ||U\Lambda_{\Omega_r} U^T - UU^T \widetilde{\Omega_r} UU^T||_F^2 = ||\Lambda_{\Omega_r} - U^T \widetilde{\Omega_r} U||_F^2 \quad (11)$$
479 Define $B := U^T \widetilde{\Omega_r} U$. Now by using the fact that $\widetilde{\Omega_r}$ can be eigendecomposed as $\widetilde{\Omega_r} = Q\Lambda Q^T$, we
480 can further simplify (11) as:
$$||\Lambda_{\Omega_r} - U^T \widetilde{\Omega_r} U||_F^2 = \sum_{i \in [p]} (\Lambda_{\Omega_r,ii} - B_{ii})^2 + \sum_{i \neq j} B_{ij}^2 \geq \sum_{i \in [p]} (\Lambda_{\Omega_r,ii} - B_{ii})^2 \geq \sum_{i \in [p]} (\mathbb{T}_{[u,v]}(B_{ii}) - B_{ii})^2,$$
481 where the last inequality holds because $u \leq \Lambda_{\Omega_r,ii} \leq v, \forall i \in [p]$. In order to achieve the first
482 equality, $B = U^T \widetilde{\Omega_r} U$ should be a diagonal matrix, which means $U^T Q = I_p \Leftrightarrow U = Q$. In this
483 case, $\text{diag}(B) = \Lambda$. To achieve the second equality, simply let $\Lambda_{\Omega_r} = \mathbb{T}_{[u,v]}(\text{diag}(B)) = \mathbb{T}_{[u,v]}(\Lambda)$,
484 which completes the proof. ∎

Table 2: Stable rank and spectral norm on SARCOS.

|  | Stable Rank | Spectral Norm |
| --- | --- | --- |
| MTL | 4.48 | 0.96 |
| MTL-WeightDecay | 4.83 | 0.92 |
| MTL-AdaReg | **2.88** | **0.70** |

## C   More Experiments

In this section we first describe the network structures used in our main experiments and present more experimental results.

### C.1   Network Structures

**Multiclass Classification (MNIST & CIFAR10)**: We use a convolutional neural network as our baseline model. The network used in the experiment has the following structure: $\mathsf{CONV}_{5\times5\times1\times10}$-$\mathsf{CONV}_{5\times5\times10\times20}$-$\mathsf{FC}_{320\times50}$-$\mathsf{FC}_{50\times10}$. The notation $\mathsf{CONV}_{5\times5\times1\times10}$ denotes a convolutional layer with kernel size $5 \times 5$ from depth 1 to 10; the notation $\mathsf{FC}_{320\times50}$ denotes a fully connected layer with size $320 \times 50$. Similarly, CIFAR10 considers the structure: $\mathsf{CONV}_{5\times5\times3\times10}$-$\mathsf{CONV}_{5\times5\times10\times20}$-$\mathsf{FC}_{500\times500}$-$\mathsf{FC}_{500\times500}$-$\mathsf{FC}_{500\times10}$.

**Multitask Regression (SARCOS)**: The network structure is given by $\mathsf{FC}_{21\times256}$-$\mathsf{FC}_{256\times100}$-$\mathsf{FC}_{100\times7}$.

### C.2   Stable Rank and Spectral Norm on SARCOS

We also show the experimental results of stable ranks and spectral norms on the SARCOS dataset. For the SARCOS dataset, the weight matrix being regularized is of dimension $100 \times 7$. Again, we compare the results using three methods: MTL, MTL-WeightDecay and MTL-AdaReg. As can be observed from Table 2, compared with the weight decay regularization, AdaReg greatly reduces both the stable rank and the spectral norm of learned weight matrix, which also helps to explain why MTL-AdaReg generalizes better compared with MTL and MTL-WeightDecay.

### C.3   Combination

As discussed in the main text, combining the proposed AdaReg with BN can further improve the generalization performance, due to the complementary effects between these two approaches: BN helps smoothing the landscape of the loss function while AdaReg also changes the curvature via the row and column covariance matrices (see Fig. 6).

On the other hand, we do not observe significant difference when combining AdaReg with Dropout on this dataset. While we are not clear what is the exact reason for this effect, we conjecture this is due to the fact that Dropout works as a regularizer that prevents coadaptation while AdaReg instead encourages neurons to learn from each other.

### C.4   Ablations

In all the experiments, the AdaReg algorithm is performed on the softmax layer. Here, we study the effects of applying AdaReg algorithm in all CONV/FC layers, all CONV layers, all FC layers, and the last FC layer (i.e., softmax layer). We first discuss how we handle the convolutions in our AdaReg algorithm. Consider a convolutional layer with {input channel, output channel, kernel width, kernel height} being $\{a, b, k_w, k_h\}$, we vectorize the original 4-D tensor to be a 2-D matrix of size $ak_w k_h \times b$. The AdaReg algorithm can therefore be directly applied on this transformed matrix. Next, we perform the experiment on MNIST with batch size 2048 in Fig. 7. The training set size here is chosen as $\{128, 256, 512, 1024, 2048, 4096, 8192, 16384, 32768, 60000\}$.

We find that simply applying the AdaReg algorithm in the softmax layer reaches best generalization as comparing to applying AdaReg on more layers. The improvement is more obvious when the training set size is small. We argue that neural networks can be realized as a combination of a complex

(a) Batch size = 256.                    (b) Batch size = 2048.

Figure 6: Combine AdaReg with BN and Dropout on MNIST.

Figure 7: Applying AdaReg on different layers in neural networks for MNIST with batch size 2048.

nonlinear transformation (i.e., feature extraction) and a linear model (i.e., softmax layer). Since AdaReg represents a correlation learning in the weight matrix, it implies that implicit correlations of neurons can also be discovered. In the real world setting, different tasks should be correlated. Therefore, applying AdaReg in the linear model shall improve the model performance by discovering these tasks correlations. On the contrary, the nonlinear features should be decorrelated for the purpose of generalization. Hence, applying AdaReg in previous layers may lead to adversarial effect.

## C.5 Covariance matrices in the prior

One byproduct that AdaReg brings to us is the learned row and column covariance matrices, which can be used in exploratory data analysis to understand the correlations between learned features and different output tasks. To this end, we visualize both the row and column covariance matrices in Fig. 8. The two covariance matrices on the first row correspond to the ones learned on a training set with 600 instances while the two on the second row are trained with the full dataset on MNIST.

From Fig. 8 we can make the following observations: the structure of both covariance matrices become more evident when trained with larger dataset, and this is consistent with the Bayesian principle because more data provide more evidence. Second, we observe in our experiments that the variances of both matrices are small. In fact, the variance of the row covariance matrix $\Sigma_r$ achieves the lower bound limit $u$ at convergence. Lastly, comparing the row covariance matrix $\Sigma_r$ in Fig. 8 with the one computed from model weights in Fig. 5, we can see that both matrices exhibit the same correlation patterns, except that the one obtained from model weights are more evident, which is

(a) Row Cov. matrix trained on 600 instances.

(b) Column Cov. matrix trained on 600 instances.

(c) Row Cov. matrix trained on 60,000 instances.

(d) Column Cov. matrix trained on 60,000 instances.

Figure 8: Recovered row covariance matrix $\Sigma_r$ and column covariance matrix $\Sigma_c$ in the prior distribution on MNIST.

due to the fact that model weights are closer to data evidence than the row covariance matrix in the Bayesian hierarchy.

On the other hand, the column covariance matrix in Fig. 8 also exhibit rich correlations between the learned features, e.g., the neurons in the penultimate layer. Again, with more data, these patterns become more evident.

Figure 9: Explained variance of different methods on 7 regression tasks from the SARCOS dataset.