[Reviews · NeurIPS 2019]

Reviewer 1



=Statistical Strength= Throughoutt the paper, you refer to the concept of 'statistical strength' without describing what it actually means. I expect it means that if two things are correlated, you can estimate properties of them with better sample efficiency if you take this correlation into account, since you're effectively getting more data. Given that two features are correlated, optimization will be improved if you do some sort of preconditioning that accounts for this structure. In other words, given that features are correlated, you want to 'share statistical strength.' However, it is less clear to me why you want to regularize the model such that things become correlated/anti-correlated. Wouldn't you want features to be uncorrelated (covariance is the identity)? Rather than having correlated redundant features, why not just reduce the number of features, since this is a tunable hyperparameter? I can imagine an argument that it is better to have a model with lots of features where there is redundancy, and you carefully control for this redundancy instead of one where you reduce the number of features (num activations in intermediate layers). This argument would be consistent with recent work on overparamterization and implicit regularization. However, you would need to test this in your experiments. =Line 46= I don't understand this statement, and I expect other readers will not too. Why set priors that would seek to mimic the behavior of the prior-less learning method? Why would this lead to better generalization? =Effect of batch size= I was concerned by how big the effect of the batch size was in figure 2. The difference between 2B and 2A is considerable, for example. This suggests that there is considerable implicit regularization from using different batch sizes, and that the effect of this may be substantially bigger than the effect of Adareg. In particular, the difference between 2B and 2A for a given method seems to be bigger than the difference between methods within either 2A or 2B. I know there has been work on the connection between mini-batch SGD, natural gradient, and posterior inference in https://arxiv.org/abs/1806.09597 I am not up to date with this literature, and I expect there is follow-on work. It is important to comment on this. In general, assessing the impact of regularization techniques is difficult because there are so many ways you can regularize. You could also do early stopping, for example. The important regularization technnique that I wish you had discussed more is simply using a model with less capacity (fewer hidden activations). See my comment on 'statistical strength.' =Minor comments= The claims in the first sentences of the abstract are unnecessarily general. Why make broad statements about 'most previous works' or explain how all neural networks are designed? You repeatedly refer to things being 'diverse' in the intro, but don't explain or motivate what that means enough.

Reviewer 2



The paper proposed a to fit a matrix-variate normal prior with kronecker covariance matrix as an adaptive and data-dependent regularization technique. It also encourages neurons to be statistically correlated to each other. An efficient block coordinate descent (hyperparams and weights) algorithm with analytical solutions are also proposed to optimizer the framework. Empirical results are showed that the proposed method can outperform weight decay and DeCoV in terms of generalization performance. Furthermore, detailed analysis such as spectral norm and correlation are provided. The paper provides a solid justification of the mechanism how matrix-variate normal prior regularizes the weight, taking into account the local curvatures. I am convinced that the proposed method can be used as a strong regularization technique. The connection between prior and weight regularization is similar to l2 regularization and Gaussian weight prior. My main complaint is the scale of the experiments though the paper focuses on the small dataset. I expect a newly proposed regularization technique should be tested with deeper network because regularization technique can be more effective when adding to a more over-parameterized network. Also, at line 205, the paper said the regularization is placed only at the last softmax layer. I wonder whether the baseline approach such as weight decay is also placed only at the last softmax layer ( It seems the provided code only has the details of the proposed method ). If so, how does the proposed regularization compared to (weight decay or l2 regularization + BN) applied to all layers? I believe this (l2 + BN) is a more common setup. The paper didn't discuss the computational overhead over weight decay in details. I am convinced that it is efficient in the EB framework (and only applied to the last softmax layer) but it should still be slower than weight decay. It would be better to have a forward-backward time comparison on a larger network. A minor issue is that the paper didn't discuss [1] in the related work. AEB is not a novel approach and related work should be discussed in more details in the revision. Also, the proposed method has limited novelty given the fact that Kronecker structure covariance matrix as posterior has been proposed in [2]. Another minor issue is that the idea that neurons are designed to be statistically related seems to conflict Dropout, which is designed to prevent co-adoption of neurons. Can authors elaborate more on this conflict in related work? [1]: McInerney, J. (2017). An Empirical Bayes Approach to Optimizing Machine Learning Algorithms. NIPS 2017. [2]: Zhang, G., Sun, S., Duvenaud, D.K., & Grosse, R.B. (2018). Noisy Natural Gradient as Variational Inference. ICML 2018.

Reviewer 3



This paper introduces a new learning algorithm that can be used to train neural networks with adaptive regularization. The authors show that the weights in each layer are correlated and then define a prior distribution over the weight to reflect this correlation (reducing the search space basically). The authors propose to use block coordinate descent algorithm to optimize the hyper parameters of this distribution and the weight of the network. The authors show that the proposed method outperforms the existing regularizers. Overall, this paper is well-written, very novel and introduces a very practical way to adaptively regularize a neural network. The following are some minor problems: 1- The authors do not study or talk about the running time of their algorithms compared to the baselines. 2- Figure 3 shows that the proposed algorithm oscillates and it is not well behaved. It would be nice if the authors could talk about this behavior.

[Author Response · NeurIPS 2019]

We would like to thank all the valuable and constructive feedback from the reviewers. Please see the following for the response, and we will make corresponding modifications in the revised manuscript.

**[Reviewer # 1 and # 2] Remarks on Motivation/Comparison with Dropout**. We would like to point out that AdaReg does not explicitly enforce the weight matrices to be positively/negatively correlated. Instead, we capture the correlations in weight matrices/gradients by defining a prior with tunable covariance matrices. From an optimization perspective, as the reviewer pointed out, we use a preconditioning matrix to change the curvature and reduce the condition number of the optimization problem. We choose the learned precision matrices $\Omega_c, \Omega_r$ (not covariance matrices) as our preconditioning matrices. As shown in line 139-140, AdaReg encourages the covariance of the effective optimization variable $vec(W') = (\Omega_c^{1/2} \otimes \Omega_r^{1/2})vec(W)$ to be identity, which means the rows/columns of $W'$ are encouraged to be uncorrelated. We note that $W$ is the original weight matrix and $W'$ can be viewed as the transformed weight matrix in the preconditioned system. As a comparison, Dropout prevents co-adaptation by randomly inhibiting the activations. Therefore, our method is orthogonal to but not contradictory with Dropout. To verify the claim, in Fig. 6 in the Appendix, we provided experiments to combine Dropout and AdaReg. The result shows further improvement ($\sim 2\%$) compared with using only AdaReg.

**[Reviewer #1] Remarks on Statistical Strength**. We use the metaphor of *statistical strength* to refer that, by taking into account the correlations between data/gradients, we improve the effective sample size. From an optimization viewpoint, reducing the number of hidden features will not help optimization since the condition number can still be very large. To address the raised concerns, we performed additional experiments using only 15 hidden units in the last fully connected layer (the original implementation has 50 hidden units) on MNIST with batch size 256. {Regularizing_Type/Hidden_Dim} with {L2/50}, {L2/15}, {AdaReg/50}, and {AdaReg/15} are 97.53%, 96.85%, 98.27%, and 98.26%. We see that when reducing the number of hidden units, L2 incurs performance drop while AdaReg maintains similar result.

**[Reviewer # 1] Remarks on Line 46.** In linear regression (see [Chapter 1, 1]), in terms of the expected mean-squared-error, the estimator of the hyperparameters obtained by empirical Bayes strictly dominates the one obtained by MLE. Inspired by this result, we explored hyperparameter learning by empirical Bayes. The hyperparameters here correspond to the "prior over weight matrix $W$", and the empirical Bayes framework corresponds to learning parameters of the prior from the correlations in data.

**[Reviewer # 1] Remarks on Batch Size.** On the MNIST dataset, for most of the methods except AdaReg and BatchNorm, we do observe that smaller batch size leads to better generalizations. The result is consistent with the observations in [2], who empirically argue that a smaller batch size corresponds to a flatter minima in convergence. We note that AdaReg is insensitive to the choice of the batch size (consistent in both MNIST and CIFAR10). From an optimization perspective, we conjecture that the preconditioning matrices (precision matrices) found by AdaReg change the curvature of the loss landscape, and it converts sharp minima to a flatter minima in the preconditioned system.

**[Reviewer # 2 and # 3] Computational Overhead and Oscillation in Fig. 3.** In Algorithm 1, the computation overhead lies in InvThreshold operation. First, as shown in line 189-192, InvThreshold's time complexity scales sub-quadratically in terms of number of parameters in the layer. Second, in practice, due to the nature of block coordinate descent, we perform the InvThreshold at every outer loop, where each outer loop contains an inner loop of several ($n = 25/50$) epochs for updating network parameters via first-order optimization. Hence the computational overhead due to InvThreshold (line 3 and 4 in Algorithm 1) is almost negligible compared with that of updating network's parameters (line 2 in Algorithm 1). On the MNIST dataset with batch size 256, for 300 epochs, CNN takes 3103 seconds to finish and our CNN-AdaReg takes 3172 seconds to complete training.

As explained in our block coordinate descent algorithm, we update covariance matrices in the outer loop which contains $n = 25/50$ inner loops for updating the model parameters. The outerloop-innerloop update causes the oscillation in the optimization trajectories in Fig. 3. We will clarfify this in the revised paper.

**[Reviewer # 2] Remarks on the Experiments.** We conducted additional experiments with AdaReg on deep residual net (18 layers) for CIFAR-10. This model contains Batch Normalization and Weight Decay as in the original paper. The results are as follows: The performance before applying AdaReg is 93.02% while applying AdaReg gives us 95.04%. In all of our experiments, the standard regularization techniques (L2, BN, DeCov, and Dropout) are applied to all the layers. We have also provided experiments of applying AdaReg to all the layers in Fig. 7 in Appendix, where we do not observe much difference as comparing to applying it only on the last layer.

We also provide additional experiments of applying L2+BN. On MNIST with batch size 256, L2 only reaches 97.53%, L2+BN reaches 97.72%, and AdaReg reaches 98.27%. Although we do see improvements for combining L2 and BN as comparing to L2 only, AdaReg still performs the best.

[1] Large-scale Inference: empirical Bayes methods for estimation, testing and prediction. Efron, Bradley, 2012.
[2] On Large-Batch Training for Deep Learning: Generalization Gap and Sharp Minima. Keskar et al., ICLR 2017.


[Meta-Review · NeurIPS 2019]

Borderline paper leaning to accept. All reviewers liked the paper but even after rebuttal have a minor concern regarding originality.